# The Multiple Interactions of RUNX with the Hippo–YAP Pathway

**DOI:** 10.3390/cells10112925

**Published:** 2021-10-28

**Authors:** Linda Shyue Huey Chuang, Yoshiaki Ito

**Affiliations:** NUS Centre for Cancer Research, Cancer Science Institute of Singapore, Yong Loo Lin School of Medicine, National University of Singapore, 14 Medical Drive, #12-01, Singapore 117599, Singapore

**Keywords:** RUNX1, RUNX2, RUNX3, YAP, Hippo, TAZ, cancer

## Abstract

The Hippo–YAP signaling pathway serves roles in cell proliferation, stem cell renewal/maintenance, differentiation and apoptosis. Many of its functions are central to early development, adult tissue repair/regeneration and not surprisingly, tumorigenesis and metastasis. The Hippo pathway represses the activity of YAP and paralog TAZ by modulating cell proliferation and promoting differentiation to maintain tissue homeostasis and proper organ size. Similarly, master regulators of development RUNX transcription factors have been shown to play critical roles in proliferation, differentiation, apoptosis and cell fate determination. In this review, we discuss the multiple interactions of RUNX with the Hippo–YAP pathway, their shared collaborators in Wnt, TGFβ, MYC and RB pathways, and their overlapping functions in development and tumorigenesis.

## 1. Introduction

RUNX transcription factors are well-established master regulators of development. They play critical roles in directing cell proliferation, lineage specification and differentiation. The three mammalian RUNX paralogs RUNX1, RUNX2 and RUNX3 have been shown to regulate hematopoiesis, bone formation and neuronal development, respectively [1,2,3,4,5,6,7]. RUNX proteins are strongly influenced by interacting proteins, which may enhance their transcriptional activities or toggle RUNX’s properties between transcription activation and inhibition. The first indication that RUNX contributes to the Hippo pathway came from an early 1999 report showing the physical interaction of all human RUNX proteins with the transcriptional coactivator Yes-associated protein 1 (YAP), and the resultant strong enhancement of RUNX’s transactivation ability [8]. This interaction is mediated by the PPxY sequence (also known as the PY motif), which is evolutionarily conserved in the C-terminal domain of the mammalian RUNX (Figure 1), to the WW domain of the YAP protein. Since then, YAP has been recognized as a key downstream effector of the Hippo pathway, an attractive target for cancer therapy as well as a promising option for regenerative medicine [9,10,11]. 

First discovered as a key regulator of organ size, the Hippo signaling cascade has now been mechanistically linked to cell proliferation, tissue homeostasis, differentiation and apoptosis [12]. The Hippo pathway is remarkable for its numerous components of WW domain proteins (e.g., YAP, TAZ and SAV1) and PY motif-containing proteins (e.g., LATS1/2 and AMOTs). WW domains may occur in isolation or in tandem, which could bestow a strong and specific binding to PY motifs [13]. Furthermore, tyrosine phosphorylation of the WW proteins and the PY motif have been shown to regulate binding affinity [14]. In mammals, the core components of the Hippo pathway are the STE20-like kinases MST1/2, the SAV1 and MOB1 scaffold/adaptor proteins, which regulate MST1/2 activities on the large tumor suppressor 1 and 2 (LATS1/2) kinases, and the transcription factor complexes comprising coactivators YAP or TAZ and their main DNA-binding partners TEAD1–4 [12]. Paralogs YAP and TAZ serve functionally redundant as well as nonoverlapping roles in transcription coactivation [15]. They are structurally similar, sharing about 60% amino acid sequence homology [15]. YAP appears to exert a stronger influence than TAZ on cellular physiology in HEK293A cells (e.g., proliferation and migration) [15]. The properties of YAP are diversified by multiple mRNA splice variants. Eight YAP protein isoforms have been characterized—the main differences are the presence of one or two WW domains, and the presence of an intact or disrupted leucine zipper domain, which mediates interaction with TAZ and pro-oncogenic phosphatase SHP2 [16,17,18]. The retention of the leucine zipper was shown to be influenced by oncogenic KRAS regulation of splicing factor SRSF3 [16]. While four of the YAP isoforms possess one WW domain, the other four possess two. These differences afforded the isoforms distinct interacting partners and transcriptional activities in regulating cell proliferation, differentiation and oncogenic potential. YAP isoforms are differentially expressed based on cell type, as well as levels of pluripotency and differentiation [17]. TAZ possesses one WW domain [15]. Moreover, TAZ contains an additional phosphodegron, indicating differences in regulation of protein stability between YAP and TAZ [19]. LATS-mediated phosphorylation of YAP and TAZ regulate YAP/TAZ subcellular localization as well as susceptibility to SCF^β-TRCP^-mediated proteosomal degradation [12]. 

YAP and TAZ function as oncogenic drivers in various solid cancers and tumor suppressors in some hematopoietic malignancies [20,21]. This review discusses the collaboration between RUNX and various components of the Hippo pathway in cell fate decisions such as proliferation, regeneration and differentiation. 

## 2. Overview of RUNX in Cancer 

RUNX proteins can function as tumor suppressor or oncogene, depending on cell context [22]. While their evolutionarily conserved DNA binding domain (also known as Runt domain) binds to the consensus RUNX motif 5′-ACCRCA-3′, the Runt domains of different paralogs showed different affinities for the motif [23]. Moreover, although the divergent C-termini of the three mammalian RUNX paralogs allow for distinct interactomes and functions, there are some structural similarities such as the transactivation and inhibitory domains, as well as the PY and VWRPY motifs (Figure 1). The VWRPY motif is necessary for RUNX interaction with transcriptional corepressors, including TLE1 [24]. RUNX can act as transcription activator or repressor, depending on interacting proteins and post-translational modifications [24]. RUNX proteins interact with a medley of transcription regulators/chromatin modifiers, which include coactivators and corepressors. RUNX1, 2 and 3 have been reported to interact with acetyltransferase p300, histone deacetylases (HDAC) and corepressor mSin3A; RUNX1 and 3 bind to histone methyltransferase SUV39H1; RUNX2 binds to NAD-dependent histone deacetylase and tumor suppressor SIRT6 [25,26,27,28,29,30,31]. 

*RUNX1* is a key player in definitive hematopoietic stem cell formation and is frequently mutated in leukemia [32,33]. Mutations in the Runt domain are frequently observed in acute myeloid leukemia (AML) and myelodysplastic syndromes. Recurrent mutations in *RUNX1* have been observed in estrogen receptor-positive luminal breast cancer, which, similarly to leukemia, may be considered as a stem-cell disorder [34,35,36]. Conversely, *RUNX1* serves as a key component of the core transcriptional TAL1–GATA3–RUNX1 complex to support the malignant state of human T cell acute lymphoblastic leukemia (T-ALL) [37]. Dominant oncogenicity for all mouse *RUNX* genes was earlier shown using retroviral mutagenesis in *CD2-MYC* mice. Retroviral insertions that led to RUNX1, RUNX2 and RUNX3 overexpression were frequently observed in virus-accelerated lymphomas [24]. 

*RUNX2* is the master regulator of osteogenic development—it regulates the osteoprogenitor proliferation and osteoblast differentiation. Heterozygous mutations in the Runt domain of *RUNX2* play a causal role in cleidocranial dysplasia, an autosomal dominant heritable skeletal disease [33,38]. Of note, *RUNX2* expression is frequently elevated in osteosarcoma [39]. *RUNX2* establishes osteoblasts in a terminally differentiated state through cooperation with retinoblastoma tumor suppressor protein (pRb) and cyclin-dependent kinase inhibitor p27^KIP1^, and disruption of this cooperation is associated with dedifferentiation in high-grade osteosarcomas [40]. *RUNX2* has also been functionally implicated in metastatic breast and prostate cancer cell lines as well as their metastasis to the bone [41,42]. 

*RUNX3* is a versatile tumor suppressor gene that has been shown to cooperate with signaling pathways such as TGFβ and Wnt to inhibit growth [22]. The frequent hypermethylation and silencing of RUNX3 in solid tumors—including breast cancer, gastric cancer and hepatocellular carcinoma—indicates a prominent role in solid tumor suppression [22,43,44]. *RUNX3*-deficient mice are associated with tumor predisposition in the gastrointestinal tract [45,46]. The stomach of *RUNX3* knockout mice exhibited reduced chief cell population, indicating differentiation defects [47]. Using an oncogenic *K-ras* lung cancer mouse model system, Lee et al. observed that RUNX3 inactivation is a key early event during lung adenocarcinoma development [48]. Mechanistically, RUNX3 interacts with TGFβ effectors SMAD2/3 to induce the transcription of cell cycle inhibitor *CDKN1A* (also known as *p21^CIP1^*) and proapoptotic *BIM* genes [49,50]. RUNX3 can also attenuate the activity of Wnt effectors TCF4-β-catenin, resulting in decreased transcription of Wnt pathway genes such as *MYC* and *cyclin D1* [46]. Furthermore, RUNX3 plays a key role in the regulation of the restriction point to defend against transformation [51]. During the restriction point—when cells decide on cell fate choices such as differentiation and G1–S transition—RUNX3 transiently forms a complex with pRb and BRD2, resulting in the synergistic induction of a key regulator of the restriction point *CDKN1A* [51,52]. 

Paradoxically, RUNX3 serves oncogenic functions in ovarian cancer and natural killer/T-cell lymphoma [53,54,55]. Moreover, RUNX3 was shown to function as both tumor suppressor and tumor promoter in pancreatic ductal adenocarcinoma [56]. While RUNX3 can inhibit proliferation, highly elevated levels of RUNX3 in pancreatic ductal adenocarcinoma can direct a metastatic program to promote cell migration, invasion and distant colonization [56]. In addition, there is increasing evidence that RUNX proteins play non-transcriptional roles during DNA repair and mitosis [57,58,59]. Not surprisingly, dysregulation of RUNX genes has been heavily implicated in disease states such as cancer and autoimmune disorders. 

## 3. Overview of YAP/TAZ in Cancer

The Hippo pathway regulates the nucleocytoplasmic shuttling of YAP/TAZ. LATS-mediated phosphorylation of YAP/TAZ blocks their nuclear accumulation and activity. In the nucleus, YAP/TAZ bind to the TEAD transcription factor, which is responsible for most of the YAP/TAZ transcriptional output [60]. Overexpression of YAP induces hyperproliferation of undifferentiated stem/progenitor cells in mouse tissues such as the gastrointestinal tract, liver and skin [61,62]. Moreover, increased TAZ/YAP activity in poorly differentiated breast tumors were associated with enrichment of stem cell signature, suggesting that TAZ/YAP bestowed cancer stem cell-like properties on breast cancer cells [63]. YAP and TAZ maintain self-renewal and pluripotency in somatic stem cells [64]. The introduction of YAP and TAZ into terminally differentiated cells can induce reprogramming into a stem/progenitor cell-like state [64]. Therefore, abnormally elevated YAP/TAZ activity and the subsequent enhancement of stem-like properties might promote tumorigenesis. 

Activation of YAP/TAZ is a key characteristic of many human cancers [65]. It was suggested that increased YAP expression might be a common event in the development of solid tumors such as colonic adenocarcinoma, lung adenocarcinoma and ovarian serous cystadenocarcinoma [66]. Moreover, YAP and TAZ are important contributors to tissue repair following injury [65]. For example, the gp130–Src–YAP signaling module serves as a critical link between inflammation and epithelial regeneration after injury [67]. The injury-related activation of YAP and the YAP-dependent inflammation response indicate that YAP is a common denominator driving proliferation in both epithelial tissue repair and cancer [65].

To promote transcription for oncogenic growth, YAP and TAZ collaborate with TEAD DNA binding proteins and activator protein-1 (AP-1, dimeric complex comprising JUN and FOS proteins) [68]. YAP/TAZ, TEAD and AP-1 form a complex at enhancers that harbor TEAD and AP-1 motifs to synergistically activate genes involved in the control of S-phase entry and mitosis [68]. Interestingly, this work also revealed a low but significant enrichment of RUNX motifs at YAP/TAZ peaks [68]. Earlier, RUNX1 and 2 were shown to interact with AP-1; RUNX2 cooperatively bound to AP-1 to activate the collagenase-3 promoter (Figure 1) [69]. Whether AP-1 binds to RUNX and YAP/TEAD simultaneously or in a mutually exclusive manner remains unclear. Moreover, YAP physically interacts with BET (bromodomain and extra-terminal) transcriptional coactivators BRD2 and BRD4 [70]. YAP, TAZ, TEAD1 and BRD4 are found in a multiprotein nuclear complex. YAP/TAZ recruits BRD4 to enhancers of growth-related genes to boost their expression, thereby mediating transcriptional addiction in cancer cells [70]. Small molecule BET inhibitors were able to mediate regression of YAP/TAZ-addicted neoplastic lesions [70]. RUNX3 has been shown to interact with BRD2. It would be interesting to examine whether the YAP–TEAD–BRD4 multiprotein complex includes RUNX proteins. 

MYC and YAP–TEAD cooperate to regulate proliferation-related genes, such as those essential for cell cycle entry, organ growth, and tumorigenesis [71]. Activation of MYC results in its extensive association with genomic sites, most of which were already occupied by TEAD [71]. Subsequent recruitment of YAP to MYC–TEAD-occupied promoters requires pre-bound MYC and is followed by full transcriptional activation [71]. YAP/TAZ activation was reported to play a crucial role in the initiation of gastric cancer, both in mouse and human. In mice, *Lgr5*-targeted YAP/TAZ activation—via conditional knockouts of *LATS1* and *LATS2*—in pyloric stem cells induced dysplastic changes and, in time, neoplasia in the pyloric epithelia [72]. MYC was also identified to be a downstream target of YAP via both transcriptional and post-transcriptional regulation [72]. Moreover, a significant correlation between *YAP* and *MYC* expression was also observed in human gastric cancer [72]. As discussed earlier, *RUNX* genes have been shown to be collaborating oncogenes in *MYC*-driven lymphoma mouse models [24]. It remains to be seen whether YAP, MYC and RUNX cooperate in tumorigenesis.

Although YAP and TAZ are well known oncogenes, both can serve as tumor suppressors in multiple cancer types [73]. YAP is absent in hematopoietic cancers, which likely reflects its anticancer function in hematopoietic cells. Ectopic expression of YAP in multiple myeloma triggered p73-mediated apoptosis after DNA damage [20]. Pearson et al. (2021) categorized solid cancers into YAP^on^ and YAP^off^ groups, where YAP serves pro- or anticancer functions, respectively. The YAP^on^ group, typified by YAP expression and wild-type RB1 expression, comprises adenocarcinomas [73]. The YAP^off^ group, where YAP is silenced, comprises small cell/neural/neuroendocrine cancers that are enriched for *RB1*^−/−^ [73]. This seminal work indicates that YAP/TAZ silencing is a key factor as to why certain *RB1*^−/−^ cells are more susceptible to transformation than others [73]. It remains unclear why TEAD transcriptional complexes occupy different enhancers in YAP^off^ and YAP^on^ cancers [73]. Interestingly, while YAP^on^ enhancers contained AP-1, FOXM1 and RUNX transcription factor motifs, these motifs were absent in YAP^off^ enhancers [73]. Instead, YAP^off^ enhancers showed enrichment in motifs for lineage-determining basic helix-loop-helix and homeobox transcription factors (e.g., *ASCL1*, *NEUROD1* and *OTX2*) [73]. It is possible, therefore, that the various transcription factors, including RUNX, compete for occupancy and contribute to the differential enhancer occupancy and activity in relation to *RB1* status. 

YAP-TEAD activity promotes tumor properties such as proliferation, migration, and invasion to play a causal role in metastasis in breast cancer and melanoma [74]. YAP/TAZ activity was increased in metastatic breast cancer when compared with nonmetastatic breast cancer tissue [63]. The leukemia inhibitory factor receptor (LIFR) is a breast cancer metastasis suppressor that functions upstream of Hippo signaling. Restoration of LIFR expression in cancer cells triggers the Hippo pathway, leading to phosphorylation, cytoplasmic localization and functional inactivation of YAP and subsequent suppression of metastasis. On the other hand, a loss of *LIFR* in nonmetastatic breast cancer cells results in the activation of YAP, which promotes migration, invasion and metastatic colonization [75]. Of note, there are RUNX consensus binding sites in the *LIFR* promoter [76]. RUNX1 was shown to bind and activate the *LIFR* promoter in a myeloid cell line [76]. It remains to be seen whether RUNX1 regulates the *LIFR* in other cellular contexts and whether other RUNX family members regulate the *LIFR* promoter to function upstream of the Hippo–YAP pathway. 

## 4. RUNX1, YAP and TEAD

Proto-oncoprotein c-Abl has the ability to switch YAP’s role from oncogene to tumor suppressor [77]. Following DNA damage, tyrosine-phosphorylation of YAP by c-Abl increases YAP’s affinity for p73 [78]. The tyrosine-phosphorylated YAP-p73 complex is then preferentially recruited to pro-apoptotic *Bax* promoter to induce apoptosis [78]. Moreover, the tyrosine-phosphorylated YAP-p73 complex formation—mediated by the WW domain in YAP and the PY motif in p73—prevents the WW domain of E3 ligase Itch from binding p73, resulting in enhanced p73 protein stability [78]. Interestingly, the modified YAP preferentially associated with p73, when compared to RUNX1. *Itch* is a downstream transcriptional target of RUNX1 (Figure 1) [79]. Under normal conditions, YAP enhances RUNX1-mediated transcriptional activation of *Itch*, leading to p73 degradation [79]. Following DNA damage, phosphorylated YAP detaches from RUNX1, resulting in decreased Itch transcription and increased levels of p73 [79]. YAP can therefore play a different role in response to DNA damage.

*RUNX1* is necessary for regulating the balance between muscle stem cell proliferation and differentiation during muscle damage repair [80]. *RUNX1* expression is significantly increased upon muscle damage [80]. RUNX1 cooperates with transcription factors MyoD and AP-1 to drive proliferation for muscle regeneration [80]. Mice lacking muscle RUNX1 showed impaired muscle regeneration while *RUNX1*-deficient primary myoblasts underwent G1 phase arrest, followed by differentiation [80]. Interestingly, TEADs were shown to be required for normal primary myoblast differentiation and muscle regeneration [81]. In undifferentiated myoblasts, TEAD4-occupied sites were enriched in RUNX and AP-1 motifs, indicating cooperation between TEAD4, RUNX1 and AP-1 in driving proliferation (Figure 1) [81]. In differentiated cells, TEAD4 binding sites showed poor overlap with Jun, but better co-occupancy with RUNX and MyoD1/Myog [81].

## 5. RUNX2, YAP, TAZ, MST2, SAV1 and SNAIL/SLUG

The Hippo/YAP pathway is involved in the regulation of immature osteoblasts and their maturation into osteoblasts. RUNX2 is well established as an important regulator of osteoblast differentiation. YAP/TAZ can also function as transcription corepressors [82]. The interaction of osteogenic master gene RUNX2 with YAP1 in osteoblastic cells results in the suppression of RUNX2 transcriptional activity [83]. In the osteoblasts, Src and Yes tyrosine kinases phosphorylate YAP to promote YAP-RUNX2 complex formation. RUNX2 then recruits YAP to the bone-specific *osteocalcin* promoter, leading to the suppression of promoter activity [83]. The ability of YAP to modulate RUNX2 transcriptional regulation of osteoblast-related genes indicates the importance of the Src–YAP–RUNX2 axis in the regulation of osteoblast differentiation [83]. RUNX2 and YAP1 cooperate to promote transformation—coexpression of RUNX2 and YAP1 significantly increases anchorage-independent growth [84]. Overexpression of YAP1 inhibits the ability of RUNX2 to suppress the promoter of cell cycle inhibitor *p21^CIP1^* [84]. RUNX2 interacts with TAZ to regulate oncogenic soluble E-Cadherin levels and tumorsphere formation in breast cancer cells (Figure 1) [85]. RUNX2 also interacts with TAZ, which serves as transcriptional activator during RUNX2-mediated induction of *osteocalcin* gene expression [86,87]. During specification of mesenchymal stem cell fate, TAZ interacts with RUNX2 and coactivates RUNX2-dependent gene transcription to promote osteoblast differentiation while inhibiting adipocyte differentiation [87]. The phosphorylation of the tyrosine residue in the PY motif is likely to negatively impact the interaction of the PY motif with the WW domain [88]. Recently, tyrosine kinase ABL was reported to phosphorylate RUNX2 at multiple tyrosine residues, including Y412 at the PY motif. This ABL–RUNX2 interaction is necessary for the transcriptional induction of a major determinant of invasion in breast cancer, MMP13 (also known as collagenase-3) [89]. 

YAP/TAZ nucleocytoplasmic shuttling are strongly influenced by the composition of the extracellular matrix [90,91]. In stem cells, mechanical cues from the extracellular environment can instruct on the decision to maintain stemness or promote differentiation [91]. Increased matrix stiffness promotes the translocation of YAP/TAZ into the nucleus, where they can interact with transcription factors [91]. For example, the length of culture time on stiff substrates has been shown to affect the activation of YAP, TAZ and RUNX2 in mesenchymal stem cells [92]. On soft hydrogels (2 kPa), both YAP and RUNX2 were excluded from the nucleus; on 10 kPa hydrogels, YAP and RUNX2 were primarily nuclear [92]. Extended culture on stiff substrata may therefore influence stem cell fate toward osteogenic differentiation via persistent RUNX2-TAZ-YAP activation [92].

Bone-marrow-derived skeletal stem/stromal cells (SSC) are necessary for skeletal development and homeostasis. SSC can differentiate into osteoblasts, chondrocytes or adipocytes. The cooperation of Snail, Slug, YAP/TAZ/TEAD and RUNX2 is important for SSC homeostasis and osteogenesis [93]. Snail and Slug were reported to interact with YAP/TAZ to regulate SSC function [93]. Snail/Slug-deficient SSCs failed to engage differentiation programs downstream of RUNX2 [93]. Interesting, Snail and Slug were found in TAZ-Runx2 multiprotein complexes [93]. The combination of TAZ with either Snail or Slug synergistically enhances RUNX2 transcriptional activity [93]. Snail and Slug have roles in epithelial–mesenchymal transition, as well as the maintenance of the stem cell-like properties in tumor cells. It remains to be seen whether RUNX2 cooperates with Snail and Slug during tumorigenesis and metastasis.

An analysis of gastric cancer patients revealed elevated *RUNX2* expression in early cancer stages and high *RUNX2* expression correlated with poor prognosis [94]. RUNX2 was found to be involved in the maintenance of self-renewal properties and malignant potential in gastric cancer cell lines [94]. A xenograft model using primary diffuse type gastric cancer cell line XN0422 with shRNA-mediated depletion of RUNX2 expression showed a significant reduction of tumor size, when compared to control cells [94]. Ectopic expression of RUNX2 in gastric cancer cell line MGC803 correlated with increased YAP expression, while depletion of RUNX2 in XN0422 reduced the YAP mRNA expression [94].

Aside from YAP and TAZ, the ternary complex formation of RUNX2–MST2–SAV1 has been reported [95]. The MST2–RUNX2 interaction is facilitated by the C-terminal domain containing the PY motif of RUNX2 and the WW domain of SAV1 [95]. MST2 phosphorylates mouse RUNX2 at Ser-339 (Ser-347 in human RUNX2) and Ser-370 (Ser-378 in human RUNX2), resulting in the inhibition of the RUNX2 transactivation ability in C2C12 mouse myoblast cells (Figure 1). The MST2-mediated phosphorylation and inhibition of RUNX2 activity might be important in osteoblast differentiation [95].

## 6. RUNX3, YAP, TEAD, SAV1 and MST2

The TEAD–YAP complex drives oncogenic growth in gastric epithelial cells by strong induction of proliferative genes [96]. Elevated TEAD–YAP expression correlates with poor prognosis in gastric cancer patients [96]. RUNX3 suppresses cancer growth by interacting with the TEAD–YAP complex and inhibiting its transcriptional activity in gastric cancer cell lines [96]. The Runt domain of RUNX3 binds directly to a region within the DNA recognition helix (denoted as α3 helix) of TEAD, resulting in the abrogation of TEAD’s DNA binding ability (Figure 1) [96,97]. Interestingly, RUNX proteins show different affinities for TEADs, with RUNX1 and RUNX3 binding stronger to TEADs, relative to RUNX2 [96]. 

The interactions between YAP, TEAD and RUNX were found to be sensitive to serum deprivation [98]. Serum deprivation is associated with the inactivation of RAC (member of the Rho family of small GTPases) signaling, subsequent LATS1/2 activation and YAP phosphorylation. YAP phosphorylation results in a marked reduction of YAP–TEAD4 interaction, and an increased YAP–RUNX3 interaction [98] (Figure 2). Moreover, YAP–TEAD4 complex formation mainly occurs at a low cell density, while YAP–RUNX3 interaction predominates at high cell density [98] (Figure 2). This work suggests that the YAP–TEAD4–RUNX3 ternary complex is an intermediate when YAP switches partners, for TEAD4 or RUNX3 [98]. Furthermore, it was proposed that RUNX3 suppresses growth in gastric cancer cells by changing the partner of YAP from TEAD4 to RUNX3 [98].

Aside from YAP–TEAD, RUNX3 also interacts with SAV1 in a MST2-dependent manner [99]. Similar to its interaction with TEAD, the Runt domain of RUNX3 is essential for interaction with the first WW domain of SAV1; the PY motif of RUNX3 is not required for interaction (Figure 1) [99]. Through SAV1, MST2 interacts and colocalizes with RUNX3 in the nucleus [99]. MST2 phosphorylates RUNX3 at residues Ser-17, Thr-69, Ser-71, Ser-77, Ser-81 and Thr-153, with the latter 4 amino acids located within the Runt domain (Figure 1) [99]. MST2 functions in conjunction with SAV1 to interfere with Smurf1-mediated RUNX3 degradation and promote RUNX3 stability [99]. MST1/2 kinases serve proapoptotic functions. Depletion of RUNX3 abrogates MST-mediated reduction of cell viability, suggesting that MST2, SAV1 and RUNX3 cooperate synergistically to promote cell death [99]. 

## 7. Convergence of RUNX, Hippo–YAP, Wnt and TGFβ Pathways

RUNX proteins physically interact with the main effectors of oncogenic signaling pathways such as Hippo, Wnt, TGFβ and pRB. The interactions of RUNX with multiple transcription factor complexes, which include YAP–TEAD [96], TCF4–β-catenin [46], SMAD2/3 [100] and pRB–E2F1 [52], potentially enable a concerted output from the different pathways for proper tissue development and homeostasis. It follows that the dysregulation of RUNX may result in aberrant outputs from multiple developmental pathways, which in turn converge to fuel oncogenic transformation.

The Hippo pathway promotes the cytoplasmic localization of YAP/TAZ, resulting in cytoplasmic sequestration of SMAD2/3 complexes and subsequent suppression of TGFβ signaling [101]. YAP/TAZ are components of the β-catenin destruction complex, which sequesters YAP and TAZ in the cytoplasm [102]. In cells with active Wnt signaling, YAP/TAZ dissociates from the destruction complex, leading to their nuclear localization and activity. The release of YAP/TAZ from the β-catenin destruction complex is important for Wnt/β-catenin signaling—the loss of YAP/TAZ is the underlying mechanism for Wnt-dependent maintenance of embryonic stem cells in an undifferentiated state [102]. Independent from its role in the β-catenin destruction complex, the Hippo–YAP signaling serves as a key downstream effector pathway of *Adenomatous polyposis coli* (also known as *APC*) [103]. YAP activation is a prevalent characteristic of tubular adenomas from patients with familial adenomatous polyposis (FAP), a cancer syndrome linked to *APC* mutation [103]. APC interacts with SAV1 and LATS1, functioning as a scaffold protein to facilitate the Hippo pathway. Indeed, a genetic analysis indicates the requirement of YAP for APC-deficient adenoma development [103]. 

The fact that RUNX proteins also interact with SAV1, β-catenin and SMAD2/3 begs the following questions: how does the presence or absence of RUNX affect this multilayered crosstalk among pathways during tumorigenesis? How does Hippo–YAP activity affect the RUNX-TGFβ target gene expression or RUNX-Wnt connection? 

## 8. Discussion

It is interesting that knockout mouse models of individual Hippo pathway genes were insufficient to induce tumor in tissues such as lung, breast and pancreas [65]. Moreover, genetic alterations of Hippo components were generally low in cancer [104]. It was proposed that YAP/TAZ’s oncogenic effects require additional events [65]. It is therefore conceivable that transcriptional cooperation of YAP/TAZ with developmental transcription factors such as RUNX modulate YAP/TAZ-driven oncogenic growth at various stages of cancer development. The ability of RUNX to interact with various components of the Hippo–YAP pathway suggests a feedback mechanism to safeguard the different stages of the Hippo signaling cascade. The activation of YAP/TAZ is strongly associated with stem-like behavior in cancer cells, proliferation, inflammation, chemoresistance and metastasis. RUNX genes are intimately involved in proliferation, stem cell regulation, and immunity [22,105]. As described above, RUNX genes have been implicated in cancer initiation, inflammation as well as metastasis. These overlaps in biological processes are indicative of the shared roles of RUNX and the Hippo–YAP pathway and the inappropriate outcomes, should either be deregulated.

Further exploring the interaction of RUNX with the YAP-Hippo pathway is likely to yield insights on regenerative medicine. As described earlier, RUNX1 is a key determinant of muscle repair [80]. Studying the synergistic effect of RUNX1 with AP-1, TEAD and YAP may promote the development of effective strategies for muscle regeneration due to severe injury or congenital muscle diseases. Moreover, the multipotential mesenchymal stem cells are frequently used for regenerative medicine. The fact that extracellular matrix stiffness modulates the activity of the RUNX2–YAP/TAZ axis in mesenchymal stem cells [92] indicates potential for bone/skeletal tissue repair.

So far, the research on RUNX interaction with the YAP–Hippo pathway has raised many exciting possibilities for stem cell studies, tissue regeneration and cancer treatment. Further in-depth studies will, in time, expand our knowledge on the regulation of stem cell fate and cancer behavior. 

## Figures and Tables

**Figure 1 cells-10-02925-f001:**
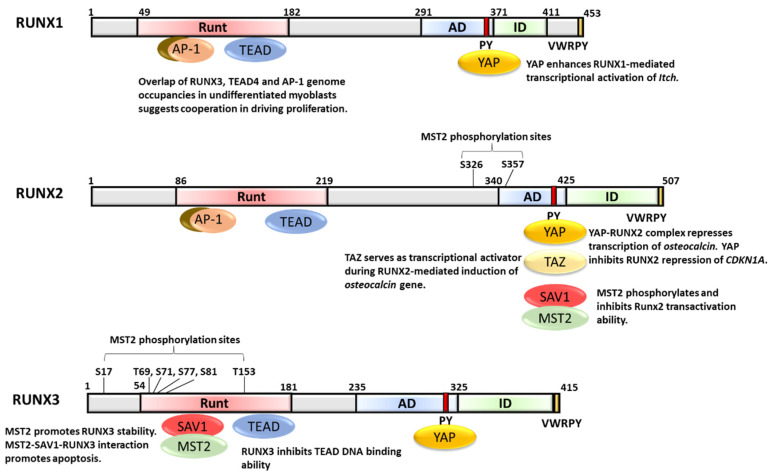
Conserved domains of human RUNX proteins and their interactions with YAP-Hippo associated proteins. The numbers indicate amino acid positions in human RUNX1 (NP_001001890.1), RUNX2 (NP_001356334.1) and RUNX3 (NP_004341.1). For RUNX2, the corresponding MST2 phosphorylation sites in the longer RUNX2 isoform NP_001019801.3 are S347 and S378 (as stated in the text). Runt, AD and ID refer to the DNA binding, activation and inhibitory domains, respectively.

**Figure 2 cells-10-02925-f002:**
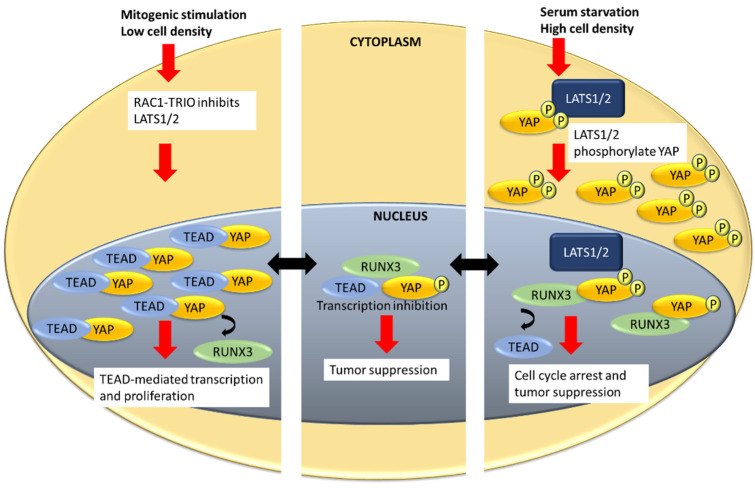
Model of RUNX3, TEAD and YAP interactions in response to serum and cell density. Mitogenic stimulation and low cell density lead to RAC1-Trio-mediated inhibition of LATS1/2. Unphosphorylated YAP accumulates in the nucleus, interacts with TEAD to the exclusion of RUNX3, to activate genes related to proliferation. Under serum starvation or high cell density, LATS1/2 phosphorylate YAP, resulting in increased cytoplasmic accumulation of phosphorylated YAP. The phosphorylated YAP in the nucleus interacts with RUNX3 and disengages from TEAD, resulting in cell cycle arrest and tumor suppression. The YAP–TEAD–RUNX3 ternary complex, which could be an intermediate between YAP–TEAD and YAP–RUNX3, inhibits TEAD-driven transcription to suppress tumorigenesis.

## Data Availability

Not applicable.

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
