# Peer review of "The Multiple Interactions of RUNX with the Hippo–YAP Pathway"

_cells, 2021, doi:10.3390/cells10112925_

Round 1

Reviewer 1 Report

The review by Chuang LS. and Ito Y. discusses the interactions between RUNX and the Hippo-YAP signaling pathway, especially focusing on the cancer. This review is well organized and comprehensively describes the important findings related the interplays between RUNX1/2/3 and Hippo-YAP pathway, and further proposes multiple potential interesting questions and points. Thus, this review is of great interest to the fields of both Hippo pathway and the RUNX family. However, some schematic diagrams are needed to make the readers easier to understand the main findings of this field.

  1. The author may draw two schematic diagrams. One figure shows the protein domains of each RUNX proteins, with the indications of the phosphorylation sites by MST2 and protein interaction interfaces with Hippo components. The other figure includes the main points from section 4-7, showing the interaction between RUNX1/2/3 with YAP/TEAD/MST2 and their response to the serum stimulation or cell density.
  2. Given the fact that three members of RUNX family all interfaces with YAP-TEAD, the author may discuss why RUNX1/2/3’s effect on YAP-TEAD is different in section 8.
  3. In the section 8, the author may summary and re-emphasize the potential future research directions which are proposed in previous sections.
  4. Line55 and line 62, it’s common that the gene function in cancers is dependent on cell context. The statement that “paradoxical roles” is not precise.

Author Response

Reviewer 1

Comments and Suggestions for Authors

The review by Chuang LS. and Ito Y. discusses the interactions between RUNX and the Hippo-YAP signaling pathway, especially focusing on the cancer. This review is well organized and comprehensively describes the important findings related the interplays between RUNX1/2/3 and Hippo-YAP pathway, and further proposes multiple potential interesting questions and points. Thus, this review is of great interest to the fields of both Hippo pathway and the RUNX family. However, some schematic diagrams are needed to make the readers easier to understand the main findings of this field.

 The author may draw two schematic diagrams. One figure shows the protein domains of each RUNX proteins, with the indications of the phosphorylation sites by MST2 and protein interaction interfaces with Hippo components. The other figure includes the main points from section 4-7, showing the interaction between RUNX1/2/3 with YAP/TEAD/MST2 and their response to the serum stimulation or cell density.

Response: We have provided 2 figures.  Figure 1 shows the structural domains of RUNX1/2/3 and their interactions with the YAP-Hippo pathway associated proteins. Protein interfaces, MST2 phosphorylation sites, as well as reported outcomes of the interactions, are as indicated.  Figure 2 shows the interaction of RUNX3 with YAP and TEAD in response to serum stimulation and cell density.  In the interest of clarity, we have omitted RUNX1/2 and MST2 from the already busy Figure 2.  We trust that the reviewer’s request for representation of MST2 has been addressed in Figure 1.

1. Given the fact that three members of RUNX family all interfaces with YAP-TEAD, the author may discuss why RUNX1/2/3’s effect on YAP-TEAD is different in section 8.

Response: There are still gaps in our understanding of RUNX1/2/3 interaction with YAP-TEAD. In Section 2, lines 73-74, we noted that the Runt domains of the different paralog show varying affinities for the consensus motif.  In the following Lines 74-76, we added new text describing that divergent C-termini of the three mammalian RUNX paralogs allow for distinct interactomes and functions.  It is likely that the differential effects of RUNX1/2/3 on YAP-TEAD stem from these 2 points.

2. In the section 8, the author may summary and re-emphasize the potential future research directions which are proposed in previous sections.

Response: We have written a paragraph lines 398 to 409 in Section 8 to re-emphasize future research directions.

3. Line55 and line 62, it’s common that the gene function in cancers is dependent on cell context. The statement that “paradoxical roles” is not precise.

Response: We have removed the “paradoxical roles” from both lines.  Please see lines 65 and 71 in the revised manuscript.

Reviewer 2 Report

The manuscript by Linda Chuang and Yoshiaki Ito is a concise summary of the current state of knowledge about links between RUNX proteins and the Hippo/YAP/TAZ tumor suppressor pathway. The work is timely and it addresses well the apparent complexity of RUNX–Hippo signaling in cancer and development and stem cells.

The work could be published as it is; however, by some additions and by eliminating several minor weak points the manuscript could be improved. The following changes are suggested: 

General Comments:

It would be important that the Authors consider composing a Figure that would depict the signaling networks and nodes of interactions among pathways discussed in the review. A sort of visual abstract that is so popular in many journals would do. We are “visual creatures” and such a general Figure would help the readers of “CELLS” a lot. An alternative could be a Table that lists the nodes of interactions between the discussed pathways.

One relatively new aspect of the Hippo YAP/TAZ pathway and RUNX signaling is the role of mechanical cues that regulate both. In line 281, this is mentioned briefly as YAP-RUNX3 complexes being dominant in high cell density. A bit more discussion about mechanobiology would be appropriate, especially knowing that Singapore is one of the world's centers of research in mechanobiology.  A review by S. M. Naqvi and L. M. McNamara In Frontiers of Bioengineering and Biotechnology 2020; doi: 10.3389/fbioe.2020.597661, could guide the Authors to several relevant publications that could be discussed.

Specific Points to Consider:

Lines 49 and 50         Please be aware that there are 8 isoforms of YAP, 4 like TAZ with one WW domain and another 4 with two WW domains.  Therefore to say that “YAP has two WW domains and TAZ has one WW domain” is not wrong but rather inaccurate. References to consider are: Gaffney, C.J., et al., Gene (2012) Volume 509, pages 215-22; Ben, C., et al., Journal of Biological Chemistry, (2020) Volume 295, pages13965-13980; Vrbsky, J., et al., (2021) Genomics, Volume113, pages 1349-1365

Line 41, 42 and Line 197, 198             There is a recent publication (He, F., et al 2021-Frontiers in Oncology, Volume 11, Article 665273), which shows that RUNX2 is phosphorylated at multiple Tyrosine residues, including Tyr 412 (the PPPY motif for YAP/TAZ binding) by the c-ABL tyrosine kinase. It is known that the decoration of Y in the PPxY motif by phosphate is known to abrogate WW domain-mediated interactions, as reviewed by Sudol, M., and Hunter, T., 2000 – Cell, Volume 103, pages 1001-1004. Perhaps these papers could be cited in the section on RUNX2 (starting at line 232)? Or just at the Introduction (lines 41-42)? The interesting thing about the recent paper is that it links Tyr phosphorylation to the expression of MMP13 and invasion in breast cancer.

Line 86 paragraph       This citation is relevant for the role of RUNX3 in tumor suppression in breast cancer: Lau Q, C., et al., 2006  - Cancer Research, Volume 66, pages 6512-6520.

Line 143 and Line 179-180     The topic in these sections is transcriptional activation versus repression. Are there any relevant citations for histone acetyltransferase (HAT, co-activators) or histone deacetylases (HDACs or SIRT, co-repressors)? It was shown that SIRT6 is a RUNX2-interacting protein that regulates glucose metabolism in breast cancer cells (Choe, M., et al 2015 Journal of Cellular Chemistry, Volume 116, pages 2210-2226). This reference could be added in the indicated sections.

Line 235          “RUNX2 also interacts with TAZ, which serves as a transcriptional activator during RUNX2-mediated induction of osteocalcin gene expression [73,74]. “

 TAZ interacts with RUNX2 to regulate E-cadherin expression and tumorsphere formation in breast cancer cells (Brusgard, J. L., et al 2015 – Volume 6(29), pages 28132-50. This reference could be also added in the indicated section.

Minor improvements to the text:

Line 24            The role of Runx2 is missing in the sentence. Because the sentence ends with “respectively” the sentence should read: “The three mammalian RUNX paralogs RUNX1, RUNX2 and RUNX3 have been shown to regulate hematopoiesis, bone formation, and neuronal development, respectively [1-7].”

Line 69            “mutations in RUNX1 have been observed in estrogen receptor-positive luminal breast cancer”

Line 245          “Interestingly, Snail and Slug were found”

Line 305           “localization of YAP/TAZ resulting in cytoplasmic”

Author Response

Reviewer 2

Comments and Suggestions for Authors

The manuscript by Linda Chuang and Yoshiaki Ito is a concise summary of the current state of knowledge about links between RUNX proteins and the Hippo/YAP/TAZ tumor suppressor pathway. The work is timely and it addresses well the apparent complexity of RUNX–Hippo signaling in cancer and development and stem cells.

 The work could be published as it is; however, by some additions and by eliminating several minor weak points the manuscript could be improved. The following changes are suggested: 

 General Comments:

 It would be important that the Authors consider composing a Figure that would depict the signaling networks and nodes of interactions among pathways discussed in the review. A sort of visual abstract that is so popular in many journals would do. We are “visual creatures” and such a general Figure would help the readers of “CELLS” a lot. An alternative could be a Table that lists the nodes of interactions between the discussed pathways.

Response: We have provided 2 figures to describe the interaction of RUNX with YAP-Hippo pathway components.

One relatively new aspect of the Hippo YAP/TAZ pathway and RUNX signaling is the role of mechanical cues that regulate both. In line 281, this is mentioned briefly as YAP-RUNX3 complexes being dominant in high cell density. A bit more discussion about mechanobiology would be appropriate, especially knowing that Singapore is one of the world's centers of research in mechanobiology.  A review by S. M. Naqvi and L. M. McNamara In Frontiers of Bioengineering and Biotechnology 2020; doi: 10.3389/fbioe.2020.597661, could guide the Authors to several relevant publications that could be discussed.

Response: We thank the reviewer for this interesting review article.   A description of how the extracellular matrix affect RUNX2, YAP, TAZ activity is provided in Section 5, lines 275 to 284.

Specific Points to Consider:

Lines 49 and 50         Please be aware that there are 8 isoforms of YAP, 4 like TAZ with one WW domain and another 4 with two WW domains.  Therefore to say that “YAP has two WW domains and TAZ has one WW domain” is not wrong but rather inaccurate. References to consider are: Gaffney, C.J., et al., Gene (2012) Volume 509, pages 215-22; Ben, C., et al., Journal of Biological Chemistry, (2020) Volume 295, pages13965-13980; Vrbsky, J., et al., (2021) Genomics, Volume113, pages 1349-1365

Response:  Thank you for the correction.  We have redressed the mistake.  Please see lines 51-60.

Line 41, 42 and Line 197, 198             There is a recent publication (He, F., et al 2021-Frontiers in Oncology, Volume 11, Article 665273), which shows that RUNX2 is phosphorylated at multiple Tyrosine residues, including Tyr 412 (the PPPY motif for YAP/TAZ binding) by the c-ABL tyrosine kinase. It is known that the decoration of Y in the PPxY motif by phosphate is known to abrogate WW domain-mediated interactions, as reviewed by Sudol, M., and Hunter, T., 2000 – Cell, Volume 103, pages 1001-1004. Perhaps these papers could be cited in the section on RUNX2 (starting at line 232)? Or just at the Introduction (lines 41-42)? The interesting thing about the recent paper is that it links Tyr phosphorylation to the expression of MMP13 and invasion in breast cancer.

Response:  Thank you for the interesting papers. We cited both.  Please see lines 269-274 in Section 5.

Line 86 paragraph       This citation is relevant for the role of RUNX3 in tumor suppression in breast cancer: Lau Q, C., et al., 2006  - Cancer Research, Volume 66, pages 6512-6520.

Response:  Thank you.  The paper is cited as Reference 44 in lines 113-116.

 Line 143 and Line 179-180     The topic in these sections is transcriptional activation versus repression. Are there any relevant citations for histone acetyltransferase (HAT, co-activators) or histone deacetylases (HDACs or SIRT, co-repressors)? It was shown that SIRT6 is a RUNX2-interacting protein that regulates glucose metabolism in breast cancer cells (Choe, M., et al 2015 Journal of Cellular Chemistry, Volume 116, pages 2210-2226). This reference could be added in the indicated sections.

Response:  A description of RUNX interaction with transcription co-activators and repressors is provided in lines 77- 85 in Section 2, with the accompanying references 25-31.  Choe et al is cited as reference 31.

 Line 235          “RUNX2 also interacts with TAZ, which serves as a transcriptional activator during RUNX2-mediated induction of osteocalcin gene expression [73,74]. “

 TAZ interacts with RUNX2 to regulate E-cadherin expression and tumorsphere formation in breast cancer cells (Brusgard, J. L., et al 2015 – Volume 6(29), pages 28132-50. This reference could be also added in the indicated section.

Response:  A description of the RUNX2-TAZ interaction and regulation of E-cadherin is provided in lines 263-265.  Brusgard et al is cited as reference 85.

 Minor improvements to the text:

Line 24            The role of Runx2 is missing in the sentence. Because the sentence ends with “respectively” the sentence should read: “The three mammalian RUNX paralogs RUNX1, RUNX2 and RUNX3 have been shown to regulate hematopoiesis, bone formation, and neuronal development, respectively [1-7].”

 Line 69            “mutations in RUNX1 have been observed in estrogen receptor-positive luminal breast cancer”

 Line 245          “Interestingly, Snail and Slug were found”

 Line 305           “localization of YAP/TAZ resulting in cytoplasmic”

Response: Thank you.  We have corrected the mistakes in Lines 24, 89, 290 and 362.

Reviewer 3 Report

Overall, it is a timely review. The RUNX family members have complicated roles including both pro-tumor and pro-apoptotic effects. The authors should overview RUNX family members and discuss their distinct mechanisms.

Author Response

Reviewer 3

Overall, it is a timely review. The RUNX family members have complicated roles including both pro-tumor and pro-apoptotic effects. The authors should overview RUNX family members and discuss their distinct mechanisms.

Response: An overview of the RUNX family members and their distinct functions have been provided.